# Effects of Different Nitrogen Fertilizer Application Rates on Soil Microbial Structure in Paddy Soil When Combined with Rice Straw Return

**DOI:** 10.3390/microorganisms13010079

**Published:** 2025-01-03

**Authors:** Xiannan Zeng, Qi Wang, Qiulai Song, Quanxi Liang, Yu Sun, Fuqiang Song

**Affiliations:** 1Engineering Research Center of Agricultural Microbiology Technology, Ministry of Education & Heilongjiang Provincial Key Laboratory of Ecological Restoration and Resource Utilization for Cold Region & Key Laboratory of Microbiology, College of Heilongjiang Province & School of Life Sciences, Heilongjiang University, Harbin 150080, China; zengxiannanzxn@163.com; 2Institute of Crop Cultivation and Tillage, Heilongjiang Academy of Agricultural Sciences, Harbin 150023, China; neauwq@163.com (Q.W.); sql142913@163.com (Q.S.); l18730852087@163.com (Q.L.)

**Keywords:** metagenomics, microbial community diversity, rice soil, soil microorganisms, straw

## Abstract

Metagenomic sequencing of the microbial soil community was used to assess the effect of various nitrogen fertilizer treatments in combination with constant rice straw return to the soil in the tiller layer of Northeast China’s black paddy soil used for rice production. Here, we investigated changes in the composition, diversity, and structure of soil microbial communities in the soil treated with four amounts of nitrogen fertilizers (53, 93, 133, and 173 kg/ha) applied to the soil under a constant straw return of 7500 kg/ha, with a control not receiving N. The relationships between soil microbial community structure and soil physical and chemical properties were determined. The results showed that the available K content of the soil significantly (*p* < 0.05) increased in soil receiving the lowest N-fertilizer dose. When applied at high amounts, N-fertilizer changed the Chao1 and ACE indices of the soil microorganisms (*p* < 0.05), and the treatments resulted in significant differences in the β-diversity of the soil microorganisms. By NMDS analysis it was demonstrated that the treatment significantly affected the structure of the soil microbial communities. Redundancy analysis showed that the main physicochemical drivers behind these differences were total nitrogen, total potassium, ammonium nitrogen, total phosphorus, and available potassium. The soil microbial communities in the control treatment were negatively correlated with nitrate and ammonium nitrogen; the lowest N-fertilizer treatment produced positive correlations with total nitrogen, total potassium, and total phosphorus and negative correlations with ammonium nitrogen; the highest dose negatively correlated with total nitrogen, available potassium, available phosphorus, total phosphorus, and pH. This study showed that moderate N fertilizer application is an effective way to increase soil microbial diversity and improve soil quality. This experiment provides technical support for the application of the alternative fertilizer technology of straw return to the field and provides a theoretical basis for rational fertilization of paddy fields in a cold climate.

## 1. Introduction

Agricultural practices in China produce large and increasing quantities of crop straw, which totaled approximately 797 million tons in 2020 and 802 million tons in 2021 (Source: National Bureau of Statistics data, http://www.stats.gov.cn (1 March 2024)). Straw is agricultural waste that is rich in carbon, nitrogen, phosphorus, and potassium, all of which are beneficial for crop growth when applied as fertilizer [1]. Agricultural practices in China currently cause soil degradation and depletion of nutrients, resulting in a decline in soil fertility that limits further agricultural developments. Such problems can be ameliorated by the application of biochar [2]. Returning crop straw to farmland is also considered a good agricultural management strategy, as it not only returns valuable nutrients to the soil, but also improves the soil aggregate structure and other properties [1,3]. Traditionally, crop straw is directly returned to the field, but this is not ideal. Straw mulching and shallow plowing results in relatively slow decomposition of the straw, leading to slow absorption and utilization of its components; the outcome is thus a relatively small improvement of soil organic matter content and less than optimal crop yields [3,4]. Since crop straw is rather bulky, burning it is a practical choice made by farmers, at a cost of air quality.

When rice straw is introduced into farmland, it undergoes decomposition facilitated by soil microorganisms, releasing essential nutrients like nitrogen (N), phosphorus (P), and potassium (K) for crop utilization [5,6]. This process helps to decrease reliance on chemical fertilizers, thereby enhancing economic benefits and improving the production-to-input ratio [7,8]. However, upon submersion in water and decomposition in the fields, rice straw may release toxins that could potentially harm rice growth. Additionally, crop straws typically exhibit a high carbon-to-nitrogen ratio, ranging from 60:1 to 80:1, which slows down the decomposition of rice residues in the soil [9,10]. Consequently, microorganisms absorb the nitrogen required for crop growth and development, leading to competition that can initially inhibit rice growth during its early stages. Nonetheless, as the rice plants continue to grow and develop, the decomposed straw releases nutrients that positively impact rice cultivation [11].

Wang et al. [12] conducted an experiment incorporating rice straw into fields under no-till conditions, revealing that, while the number of water-damaged leaves and the amount of dry matter accumulation remained unaffected, the practice extended the functional period of green leaves post-flowering and enhanced yield. Furthermore, it led to an increase in the proportion of photosynthetic product accumulation. Similarly, Meng et al. [13] observed that, although straw decomposition initially inhibits rice growth and development, it becomes beneficial in later stages. Additionally, the application of organic fertilizer to white soil rice fields significantly boosts the plants’ uptake of nitrogen, phosphorus, and potassium. Fentie et al. [14] further found that the incorporation of water hyacinth dry straw notably increased the absorption of these nutrients during rice’s maturity stage. In summary, numerous scholars have extensively investigated the impact of straw incorporation on crop growth and soil fertilization. However, there is a notable gap in systematic research exploring changes in soil physicochemical properties and microbial activity under the specific conditions of rice straw incorporation in Heilongjiang Province.

A better strategy is straw return that is prepared from pyrolysis of crop residues for use as a C-based fertilizer [15,16]. This can be applied alone or in combination with other fertilizers. Inorganic nitrogen fertilizer, in addition to a variety of other nutrients, organic matter, and trace elements, can improve the health of agricultural soil. For instance, a long-term study of corn fields fertilized with a combination of organic and inorganic fertilizers resulted in improved soil quality, as assessed by increases in soil organic matter (OM), total nitrogen (TN), available phosphorus (AP), and available potassium (AK) content, giving a long-term increase in yields [17]. Paddy fields used for double-season rice production likewise benefited from long-term application of organic fertilizer [18].

Application of fertilizer not only changes the soil’s physicochemical indicators, but also affects the abundance and distribution of soil microorganisms that have an effect on crop yield. The study of soil microorganisms in agroecosystems under different fertilization methods is key to a comprehensive understanding of soil ecology. An example is a study of winter barley that was improved by organic amendments that resulted in increased bacterial biomass, with direct effects on soil organic carbon content and C/N ratios [19]. Even if the total microbial diversity remains unchanged, application of organic fertilizers alone or in combination with inorganic fertilizers can increase crop yields by promoting the abundance of beneficial soil microorganisms [20,21]. Despite a plethora of studies, currently there is no consensus on how organic fertilizers affect soil microbial communities, as there is too much variation in the fertilizer types, plant types, and soils that have been studied.

Rice is a major food source of global importance and represents a major food crop in China [22]. A major rice-producing area is Heilongjiang Province, located in the black soil belt of North-East China [23]. Rice production results in large quantities of rice straw, which can be returned to the soil. Past studies have described the effect of rice straw return on crop yields [24,25] and soil physicochemical properties [26,27], but fewer data exist on the effects on soil microorganisms. In an attempt to close this knowledge gap, here we applied different N-fertilizer amounts under constant application of rice straw and studied the effect on the microbial community structure of soil during rice production. We based the doses of N-fertilizer on recommendations for its reduced use by the Heilongjiang Academy of Agricultural Sciences, and all test plots received equal amounts (7500 kg/ha) of straw returns. We applied genome sequencing technology to explore the effects of rice straw and fertilizer reduction on the soil microbial genes and established their relationship with soil physicochemical properties.

## 2. Materials and Methods

### 2.1. Experimental Site, Soil Properties, and Rice Variety

The location of the field investigations, which were conducted in 2017, was the Modern Agricultural Demonstration Area of the Heilongjiang Provincial Academy of Agricultural Sciences, between the cities of Harbin and Changchun in North-East China (latitude 45°49′ N, longitude 126°48′ E, altitude 117 m). The local mesothermal continental monsoon climate has short summers, an annual average temperature of 4.5 °C, and an effective cumulative temperature of up to 2700 °C. The annual growth cycle of rice crops lasts 155 d, and an average annual precipitation of 569.1 mm occurs mainly between June and September. The local soil nutrient contents in the 0–20 cm layer were characterized as 29.17 g/kg SOC, 2.81 mg/kg TN, 79.56 mg/kg AN, 55.84 mg/kg AP, and 168.42 mg/kg AK. The rice variety used in the experiment was Long Rice 21 (cultivated by the Research Institute of Cultivation of Heilongjiang Provincial Academy of Agricultural Sciences). Only one crop was produced per year. The soil type utilized for this test was black soil. The nutrient content of the 0–20 cm soil layer was as follows: the soil organic matter content was 26.50 g/kg, the total nitrogen content was 2.01 g/kg, the effective nitrogen content stood at 79.56 mg/kg, the effective phosphorus content was 55.84 mg/kg, and the available potassium content was 168.42 mg/kg.

### 2.2. Experimental Design

The area was divided into plots of 24 m^2^. Prior to the experiment, all remaining straw was removed from the soil surface. The rice straw used in the experiment was obtained from the harvest of the previous season. This straw was air-dried and cut into fragments of approximately 5 cm, to then be evenly distributed over the field and mixed into the soil with a rotary tiller. One week after application of the straw, rice plants were transplanted. For this, seeds were sown on 15 April, and the resulting rice seedlings were transplanted on 15 May in rows separated by 30 cm at a plant spacing of 13 cm, with three plants/hole. All management measures were implemented in accordance with the requirements of local conventional cultivation.

The amount of straw returned to the field (see Table 1, 7500 kg/ha) is calculated according to the actual local production needs. The grain-to-grass ratio of the straw returned to the field is an important parameter for determining the amount of rice straw returned to the field, as it helps evaluate the relationship between the yield of straw and the grain yield, and then determines the reasonable amount of straw returned to the field (in this experiment it was calculated based on a grain-to-grass ratio of 1:1).

Urea served as the nitrogen fertilizer in this study, and it was administered in three separate applications: as a basal fertilizer, a tillering fertilizer, and a greening fertilizer. The total amounts of nitrogen fertilizer (in terms of pure nitrogen) applied were as follows: (N4) 173 kg·ha, (N3) 133 kg·ha, (N2) 93 kg·ha, (N1) 53 kg·ha, and (N0) 0 kg·ha. For all treatments, 46 kg·ha of pure phosphorus and 75 kg·ha of pure potassium was applied. Diammonium phosphate was chosen as the phosphate fertilizer and was applied once as a basal fertilizer. Potassium chloride was used as the potassium fertilizer and was administered in two applications: as a basal fertilizer and as a tillering fertilizer.

A total of five randomized treatment plots were set up. One plot (S2N0) received no fertilizer and straw, whereas the other four (S2N1-S2N4) received identical rice char and P fertilizer, with variable N-fertilizer doses (Table 1). All treatments were performed with three field replications per treatment.

The research site, situated in a temperate zone characterized by a cold climate, was selected to comprehensively assess the impact of long-term straw incorporation and fertilizer application on soil properties and crop growth. To this end, soil samples were diligently collected in 2023, following six years of consistent practice. Soil samples were collected during the rice harvest in 2023. At five locations per treatment plot, distributed in an S shape, the soil was collected with a soil auger at a depth of 0–20 cm. These five samples were mixed into one composite sample per plot. After removal of stones and coarse plant residues, the soil was sifted through a 2 mm sieve. One portion was air-dried and used to measure physicochemical properties according to the literature [28], with the results summarized in Table 2. The other part was stored at −80 °C for microbial DNA extraction.

### 2.3. Extraction of Soil Microbial DNA and Metagenomic Sequencing

DNA was extracted using a Fast DNA^®^Spin Kit (MP company, National City, CA, USA) for Soil. The quality of the resulting genomic DNA was verified by 1% agarose gel electrophoresis. The DNA was fragmented to a size of approximately 400 bp using a Covaris M220 sonicator, and PE libraries were constructed using a NEXTFLEX™ Rapid DNA-Seq Kit (Revvty company, Hongkong, China). Sequencing was performed on an Illumina NovaSeq Sequencing Platform by Beijing BioMac Bioinformatics Co (Biomic company, Beijing, China).

Multiple samples were mixed in parallel for sequencing after the introduction of an index tag sequence for sample identification. Raw sequences in fastq format were subjected to quality control by Fastp (v0.20.0) to remove low-quality and ambiguous sequences. The resulting optimized sequences were spliced and assembled using Megahit (v1.1.2) software, and fragments below 300 bp were removed from the results. Contigs were determined by overlap, and open reading frames (ORFs) were predicted using Prodigal (v2.6.3) (https://github.com/hyattpd/Prodigal) (10 March 2024). The predicted genes were clustered with a similarity ≥90% and a coverage ≥90% using CD-HIT (v4.6.1) software (http://www.bioinformatics.org/cd-hit/) (10 March 2024). The longest gene in each cluster was taken as the representative sequence to construct a non-redundant gene set. The sequencing data were compared with the non-redundant gene set using SOAPaligner (v2.21) (similarity ≥ 95%), and the abundance information of the OTUs in the corresponding samples was reported.

### 2.4. Data Analysis

The difference of soil physico-chemical properties was analyzed using one-way ANOVA at a significance levels of *p* < 0.05 using SPSS statistical package (version 26.0). Diversity indices (Shannon, Simpson, Chao1, and ACE) were calculated based on standard procedures, using OTUs to generate the α-diversity indices. One-way analysis of variance (ANOVA) was applied at a significance level of *p* < 0.05 using SPSS statistical package (version SPSS 26.0), and significant differences between groups (*p* < 0.05) were determined by independent samples t-test. Non-metric multidimensional scaling (NMDS) and Principal Coordinate Analysis (PCoA) were applied, and histograms were generated of dominant species and of species abundance for each treatment. These calculations were performed using R (version 4.1.3). The significance of difference tests and linear discriminant analysis effect size (LEfSe) analyses were used to determine significant between-group differences; for this, the Maybach platform was used (www.biocloud.net). Pearson correlation analysis of soil physico-chemical properties and soil α-diversity indices were generated using the R software vegan package.

## 3. Results

### 3.1. Analysis of Soil Physico-Chemical Properties

Under the practice of straw incorporation, varying quantities of chemical fertilizers exerted a significant influence solely on soil pH, total nitrogen, total soil phosphorus, and soil available potassium (as indicated in Table 2, with *p* < 0.05). However, no notable effect was observed on other soil nutrient contents (as depicted in Table 2, with *p* > 0.05). Notably, soil total nitrogen exhibited a pattern of initial decrease followed by an increase as the application rate of chemical fertilizers diminished, achieving its peak with N4 treatment. Specifically, the total nitrogen content in the N4 treatment group surpassed that in the N0 treatment group by a significant margin of 17.61% (as shown in Table 2, *p* < 0.05). Conversely, soil total phosphorus demonstrated an initial increase and subsequent decrease with decreasing chemical fertilizer application, peaking with N2 treatment. Regarding soil available potassium content, the trend was observed to be N3 > N1 > N0 > N2 > N4, with the N4 treatment group showing a significant difference compared to the other four treatment groups.

### 3.2. Analysis of Soil Microbial Diversity

The different N-fertilizer treatments with constant rice straw application affected the alpha-diversity of the soil microorganisms (bacteria and archaea combined). Their Shannon and Simpson indices produced an increasing trend after fertilizer application (Figure 1), whereas the Chao1 and ACE indices increased with lower application (S2N1 compared to S2N0) but decreased again with higher N application (Figure 1). All indices were lowest in the control plots without N-fertilizer application (S2N0).

According to the Pearson correlation analysis, the Chao1 and Ace indices of the soil microbial community were positively correlated with soil pH (*p* < 0.05) and TP (*p* < 0.01) (Table 3), while the Shannon index of the soil microbial community was negatively correlated with soil NO_3_^−^ (*p* < 0.05).

The results of the NMDS analysis based on Bray–Curtis distances indicated that the soil microbial community structure was indeed affected by fertilizer application (Figure 2A), although there was considerable variation among the triplicates of each treatment, in particular for S2N1 and S2N2. The PCoA analysis further indicated that fertilizer application significantly affected the soil microbial β-diversity under the various treatments (Figure 3).

### 3.3. Analysis of Soil Microbial Taxa

The identified sequences were attributed to bacterial and archaeal phyla, and their distribution was compared in a bar chart (Figure 3A). In all soil samples, the phylum of Proteobacteria dominated, followed by Chloroflexi. Together, these two phyla accounted for 58% to 64% of the identified communities, and their fractions increased with N application. The relative abundance of Bacteroidetes increased at lower N application amounts compared to the control, but it decreased again in the soil receiving the highest doses. The fraction of Acidobacteria was lowest in S2N1.

Sequences were also attributed to genera, and the relative abundance of the ten most abundant genera are shown in Figure 3B. At the genus level, *Sphingomonas*, *Geobacter*, *Anaeromyxobacter*, and *Vibrio* were the four most abundant bacterial genera, in combination accounting for 57% to 74% of the most abundant genera. Of note is the strong expansion of *Vibrio* in S2N1.

Venn diagrams were used to compare the number of common and unique OTUs in the samples (Figure 4). A total of 3327 OTUs were identified, of which 2762 (83% of the total) were shared in the soil from all treatment groups (Figure 4). The total number of OTUs detected per group was highest for S2N2 and lowest for S2N0. Treatment-specific OTUs were also detected, with the highest number, 36, being found exclusively in soil from the S2N2 treatment, and the lowest number, 16, in soil from the S2N0 control.

### 3.4. Correlation Analysis

Correlations were assessed by means of RDA/CCA analysis between abundance of bacterial genera and the soil physicochemical parameters (Figure 5). The findings indicated that the main physicochemical driver responsible for the observed differences in the soil microbial community between the treatments was AN, as a direct consequence of the N-fertilizer treatments. The main genera affected by this were *Vibrio*, *Opitutus*, *Polaromonas*, *Thiobacillus*, and *Nitrospira*. Correlations were also determined between the soil physicochemical parameters and the complete microbial community of the different treated soils (Figure 6). The soil microbial community of S2N1 positively correlated with TN, TP, pH, AP, and AK, but it correlated negatively with NH_4_^+^. For S2N2, the soil microbial community positively correlated with AN, but negatively correlated with NO_3_^−^ and SOC. The soil microbial communities of both S2N3 and S2N4 positively correlated with NH4^+^ and NO_3_^−^.

### 3.5. Analysis of Significance of Differences Between Groups

LEfSe analysis of the sequences obtained from all groups identified 16 biomarkers. There was a single biomarker, of the phylum Proteobacteria, identified for the S2N3 group (Figure 7). The phylum Betaproteobacteria, its order Nitrosomonadales, and the genera of *Sideroxydans* and *Sulfuritalea*, including *S. hydrogenivorans* therein, as well as *Sulfuricurvum*, were the main biomarkers in the S2N4 group, indicating that Betaproteobacteria had the most significant effect following high N-fertilizer treatment. In contrast, for the S2N1 treatment group, *Archangium*, Deltaproteobacteria, Bacteroidetes, and Syntrophobacterales were the main biomarkers, with *Archangium gephyra* being the most significant contributor to the differential effect. Biomarkers for S2N2 were identified as *Lysobacter* and Desulfobacterales, and the two biomarkers for S2N0 were Nitrospira and Anaeromyxobacteraceae.

## 4. Discussion

### 4.1. Soil Microbial Diversity Was Changed by Nitrogen Fertilizer Application Under Straw Return

We report that application of a high amount of fertilizer combined with rice straw return significantly decreased the alpha-diversity of soil microorganisms. When N-fertilizer was applied at a low amount in combination with rice straw return, an increased Shannon index of soil microorganisms was observed, together with decreased Chao1 and ACE indices, while these treatments did not affect the Simpson index (Figure 1). The results of the NMDS analyses showed that the amount of N-fertilizer under the tested conditions significantly affected the beta-diversity of soil microorganisms. This is because, on the one hand, moderate application of N-fertilizer can promote microbial activity. At moderate doses, N-fertilizer provides additional N sources for soil microorganisms and promote their metabolic activities and growth, slightly changing the soil ecological environment and enriching the diversity of microbial communities. This is consistent with the results of previous studies [29,30]. On the other hand, some studies have shown that the addition of fertilizer alone significantly altered the diversity of soil microorganisms, with a small amount increasing and larger amounts suppressing the diversity of soil microorganisms. However, in our work we combined fertilizer addition with straw return; straw application resulted in changes in both the content and type of organic carbon in the soil, which counteracted the changes in stoichiometry between soil nutrients caused by fertilizer addition, and thus the changes in indicators for assessing soil microbial diversity were more complicated. It has been shown that straw return to the field can significantly increase the Chao1 and Shannon indices of soil microbial communities [31], which suggests that the soil microbial response to straw return differs from that to chemical fertilizer.

Other parameters such as the quality of straw, soil type, and so on further affect the outcome. Straw return increases the organic matter content of the soil, enriches the carbon source in the soil, and provides diverse organic matter and nutrients for microorganisms [32,33]. Straw can also improve the soil structure, increase the number of aggregates, increase the permeability and air content of the soil, and provide suitable space for microbial growth [34]. In conclusion, soil microbial diversity and richness can be maintained with little or no nitrogen fertilizer applied on the basis of straw fertilization.

### 4.2. Effects of Nitrogen Fertilizer Application on Soil Microbial Composition Under Straw Return

The results of our study showed that the soil samples from the low application of fertilizer (S2N2) treatment group had the highest number of unique OTUs (Figure 4), suggesting that the dominant species were the most abundant in the treatment with low application of nitrogen fertilizer. However, the relative abundance of Rhodobacteracea and Bradyrhizobieceae families was significantly lower in the SN1 treatment group relative to the other two treatment groups (Figure 2). This result suggests that, although low amounts of N fertilizer application increased soil microbial diversity, they also changed the structure of soil microbial communities and reduced the abundance of dominant species. This unique change in soil microbial composition under simultaneous N-fertilizer and straw return conditions may be attributed to alteration of soil enzyme activities. Straw return can promote soil microbial activity and thus have an important regulatory effect on soil enzyme activity. Soil enzymes are indicators of soil health and play an important role in soil nutrient cycling in agroecosystems [35]. It was found that application of straw and nitrogen fertilizer increased β-glucosidase, β-xylosidase, and N-acetylglucosaminidase activities [35]. It was also reported that soil urease and acid phosphatase activities were higher in the 0–20 cm soil layer when straw was applied [36]. Mixing straw into deeper soils for plowing accelerates nutrient cycling in the subsoil layer, thereby increasing soil enzyme activities. Yet another study showed that returning straw to the field by rotary plowing significantly increased soil enzyme activities at 0–10 cm [37]. The application of nitrogen fertilizer under straw return can regulate the secretion of soil enzymes, influence soil enzyme activities, and thus change the composition of soil microbial communities. In future studies, we will investigate the effects of straw return and nitrogen fertilizer application on the environment, crop yields, and soil microbiota in relation to soil enzyme activities and crop metabolism.

### 4.3. Effects of Nitrogen Fertilizer Application on Soil Fertility Under Straw Return to Field

We observed that the application of a low amount of nitrogen fertilizer combined with straw return resulted in a significant increase in AK content in the soil, but this was not observed for the other application rates; the other physicochemical parameters of the soil were not strongly affected (Table 2). The microbiota in the soil positively correlated with AK (Figure 6). Moreover, application of crop residues affects soil N, P, and K content, and straw return is an effective way to reduce soil N, P, and K loss, but its benefits vary depending on soil texture [38]. A field micrograph experimental study showed that straw return to the field can improve nitrogen utilization efficiency [39]. It is likely that the differences in soil texture and the dominant microorganisms in the soil may have led to significant variations in soil AK content. This form of K can be directly absorbed and utilized by plants, and AK content is beneficial for crop growth, development, and yield [38]. The highest soil value of quick-acting potassium was reached in the treatment group with the smallest amount of applied N-fertilizer. On the one hand, straw return improves the adsorption of potassium by increasing soil humic acid in large aggregates; on the other hand, it can improve the effectiveness of phosphorus and potassium in the soil to a certain extent, so that it is easier to be absorbed by plants [40]. In conclusion, attention should be paid to the appropriate application of nitrogen fertilizer when combined with straw return in order to avoid the negative impact caused by excessive nitrogen fertilizer and to realize the optimal level of soil nutrients and pH.

In summary, when combined with straw return to the field, a small amount of N-fertilizer application can improve the diversity of soil microorganisms, promote advantageous microorganisms that potentially function in the degradation of pollution, promote plant growth, and improve nitrogen cycling, in addition to improving the soil content of phosphorus and potassium. This experiment provides technical support for the application of fertilizer application technology under straw return and provides a theoretical basis for the sustainable development of agriculture with reduced fertilizer application. The mechanism of action how fertilizer application and straw return affect the microbial diversity and community structure in the studied soil needs further in-depth studies.

### 4.4. Effects of Nitrogen Fertilizer Application on Soil Alpha Diversity Under Rice Straw Return Conditions

This study revealed that the application of chemical fertilizers under rice return significantly impacts bacterial alpha-diversity (Figure 1). Specifically, the bacterial diversity in soil treated with a small amount of nitrogen fertilizer (53 kg/hm^2^) or no nitrogen fertilizer (0 kg) was significantly higher compared to other chemical fertilizer addition groups (Figure 1). This outcome can be attributed to the fact that appropriate nitrogen fertilizer application plays a crucial role in promoting microbial activity. A modest amount of nitrogen fertilizer not only provides additional nitrogen sources for soil microorganisms [41,42], fostering their metabolic activities and growth [43], but also alters the soil’s ecological environment, stimulating the growth and diversity of certain specific microbial species, thereby enriching the soil microbial community. These findings align with previous research results [44,45].

The results indicate that the change in microbial diversity is correlated with soil pH, TP, and NO_3_^−^ (Table 2). This is consistent with the results of Kaitlin et al. [46]. Prior studies have shown that the sole addition of chemical fertilizers can significantly alter soil fungal diversity [47,48]. A small amount of fertilizer can increase microbial diversity, whereas excessive application can markedly inhibit it [49]. In this study, however, since straw was incorporated into the field concurrently with the addition of chemical fertilizers, the changes in organic carbon content and types induced by straw incorporation counterbalanced the stoichiometric alterations in soil nutrients caused by the fertilizers. Consequently, significant changes were observed in soil bacteria. A previous study demonstrated that the combined application of organic fertilizer and straw incorporation significantly enhances fungal diversity in black soil [50]. This suggests that soil microorganisms respond differently to straw incorporation and chemical fertilizer addition, which is also closely linked to factors such as straw quality and soil type. Straw incorporation increases the soil’s organic matter content, enriching its carbon sources and providing a variety of organic matter and nutrients for microorganisms. Furthermore, it improves soil structure, increases aggregate numbers, enhances soil permeability and air content, and creates a favorable environment for microbial growth. In summary, on the basis of straw incorporation, only a minimal or no application of nitrogen fertilizer is required to maintain the diversity and abundance of microorganisms in the soil.

## 5. Conclusions

The effects of different amounts of nitrogen fertilizer, applied in combination with a constant amount of rice straw return, on soil microbial diversity and community structure were determined. A significant effect on the amount of available potassium in the soil was observed for the lowest administered amount of N-fertilizer. The highest amount of N=fertilizer application affected the microbial Chao1 and ACE indices, and all three treatments resulted in significant differences in the β-diversity of the soil microorganisms. NMDS analysis confirmed that the tested treatment had a significant effect on the structure of the soil microbial community. The main physicochemical drivers of soil microbial community differences among the three different fertilizer application treatments were TN, TK, NH_4_^+^, TP, and AK. We demonstrated that the structure and diversity of soil microorganism communities was altered by the treatments. We conclude that the application of appropriate amounts of nitrogen fertilizer is an effective way to increase soil microbial diversity and improve soil quality. This practice has a favorable effect on preserving the health and stability of the soil.

## Figures and Tables

**Figure 1 microorganisms-13-00079-f001:**
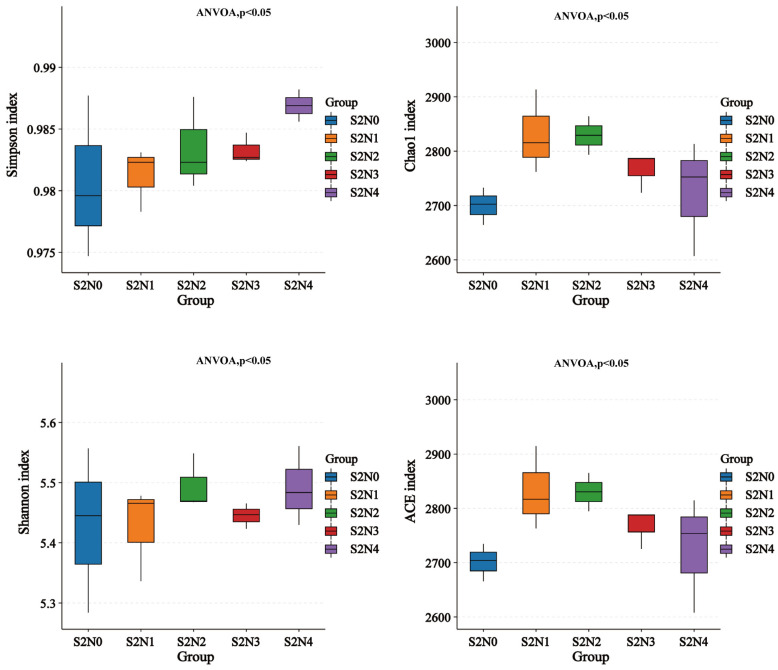
Diversity indices of microorganisms in the soil following different treatments. Statistical significance among different treatments was analyzed using ANOVA and Tukey’s post-hoc test.

**Figure 2 microorganisms-13-00079-f002:**
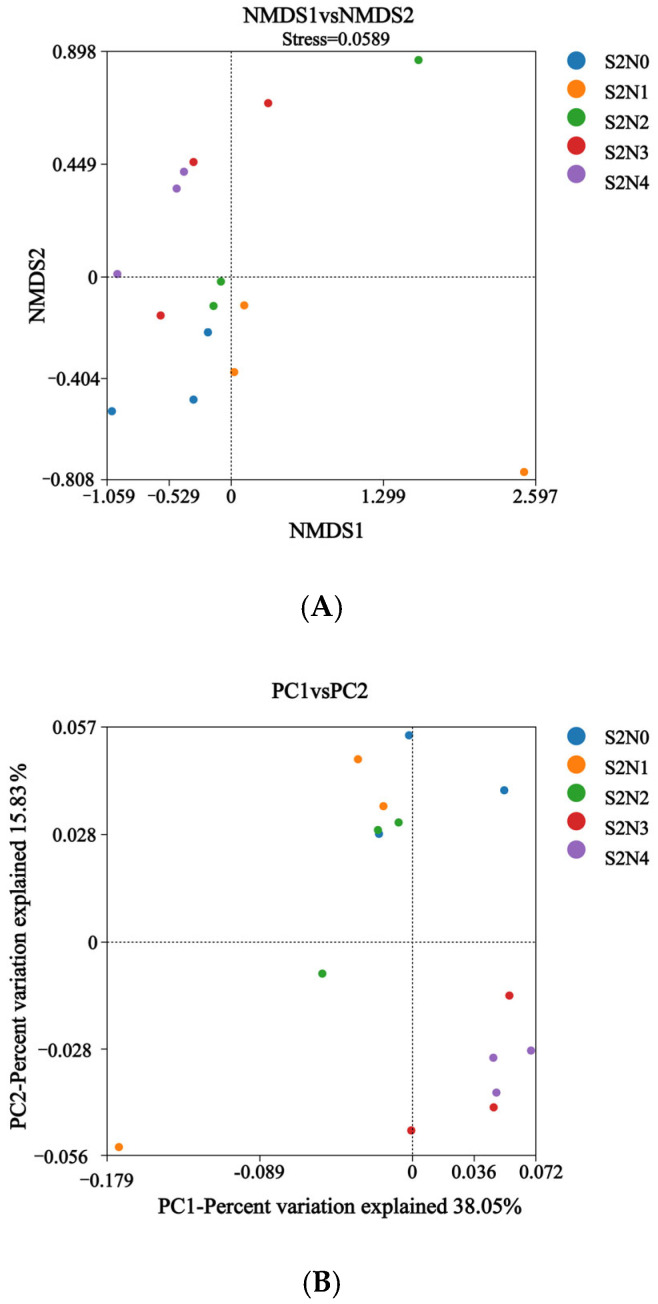
NMDS analysis (**A**) and PCoA analysis based on Bray–Curtis distance (**B**) of the soil microbial communities.

**Figure 3 microorganisms-13-00079-f003:**
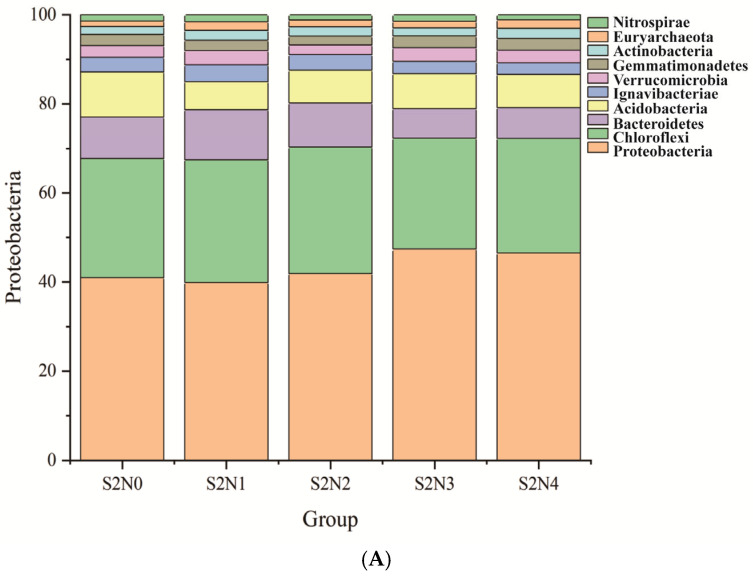
Relative abundance of the top 10 microbial phyla, including 9 bacterial and 1 archaeal phyla (**A**), and of the top 10 bacterial genera (**B**) in the soils after the different treatments.

**Figure 4 microorganisms-13-00079-f004:**
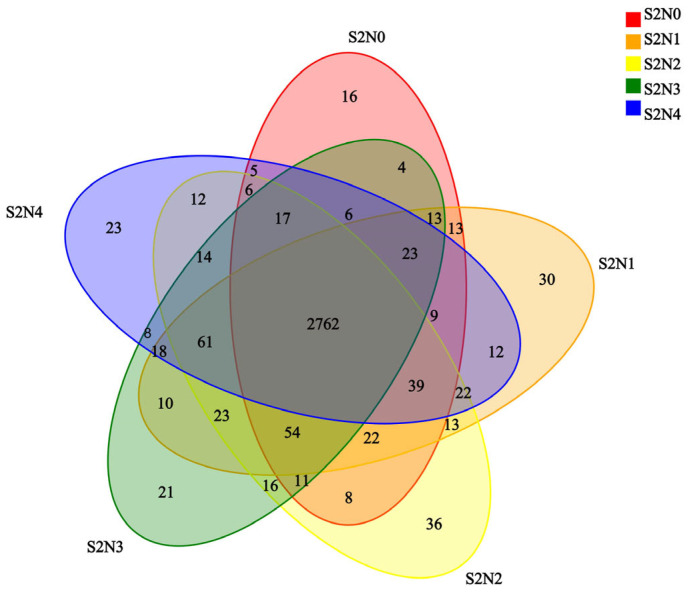
Venn diagram showing the number of common and shared OTUs of the microbial communities within the five treatment groups.

**Figure 5 microorganisms-13-00079-f005:**
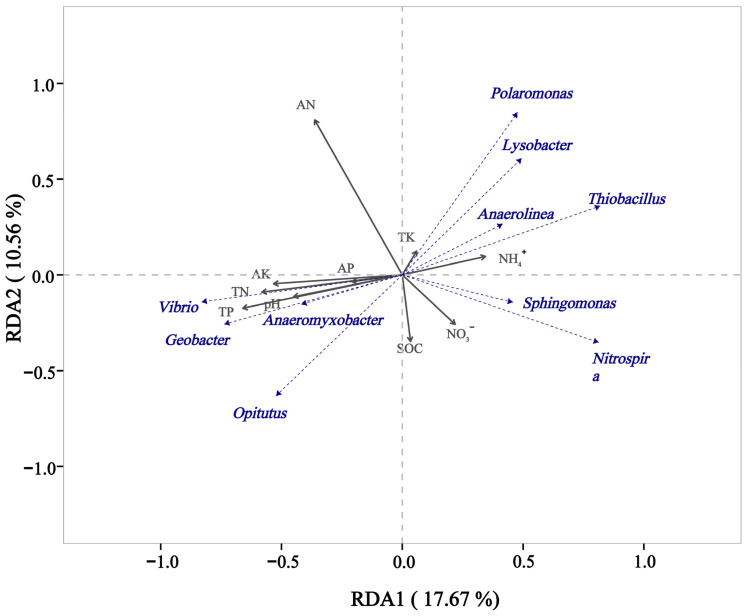
RDA/CCA analysis showing correlations between bacterial genera and the soil physicochemical properties.

**Figure 6 microorganisms-13-00079-f006:**
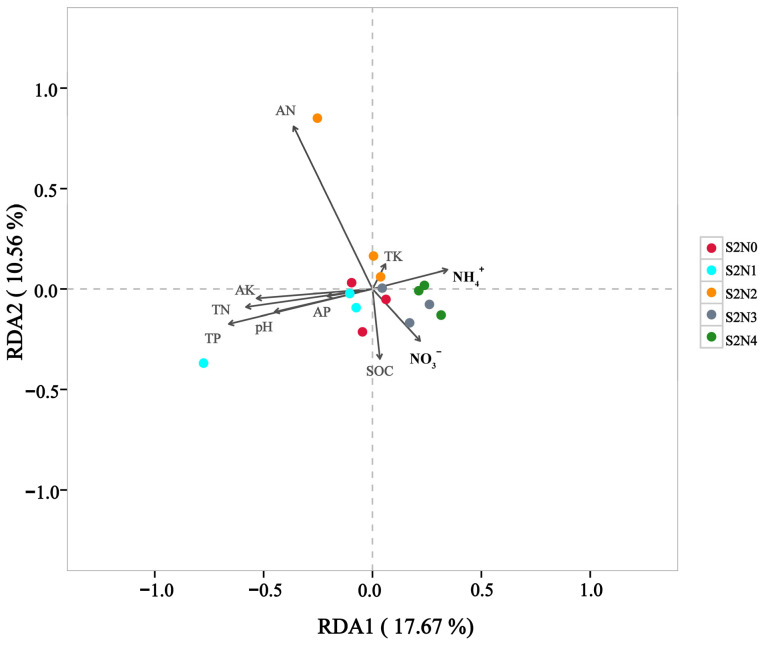
RDA/CCA analysis showing correlations between total microbial communities of the treatments (with triplicates shown) and the soil physicochemical properties.

**Figure 7 microorganisms-13-00079-f007:**
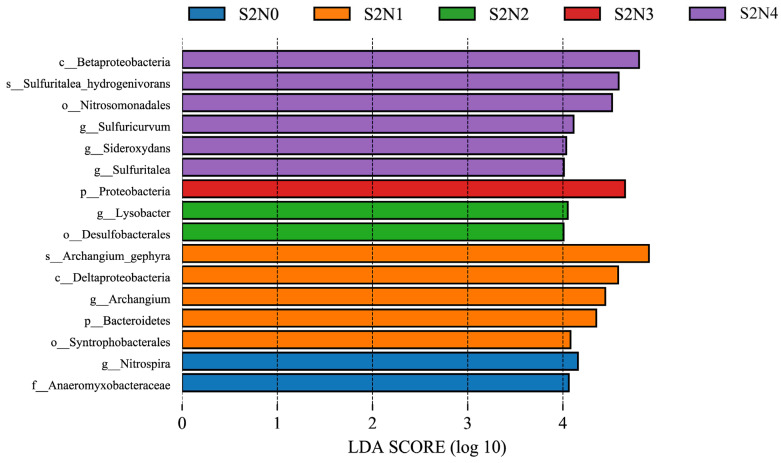
Results of LEfSe analysis of the soil microbiota after the five treatments.

**Table 1 microorganisms-13-00079-t001:** Straw return rate and fertilization treatments.

Treatment	Amount of Straw Returned to the Field (kg/ha)	Nitrogen Fertilizer Usage (Pure Nitrogen kg/ha)				Phosphorus Fertilizer Usage (Pure Phosphorus kg/ha)	Potassium Fertilizer Usage (Pure Potassium kg/ha)	
		Base Fertilizer	Return Fertilizer	Tiller Fertilizer	Total Nitrogen	Base Fertilizer	Base Fertilizer	Tiller Fertilizer
S2N4	7500	86.5	51.9	34.6	173	46	37.5	37.5
S2N3	7500	66.5	39.9	26.6	133	46	37.5	37.5
S2N2	7500	46.5	27.9	18.6	93	46	37.5	37.5
S2N1	7500	26.5	15.9	10.6	53	46	37.5	37.5
S2N0	7500	0	0	0	0	46	37.5	37.5

**Table 2 microorganisms-13-00079-t002:** Soil chemical properties under each treatment.

Treatment	pH	AN (mg/kg)	AP (mg/kg)	AK (mg/kg)	SOC (g/kg)	TN (g/kg)	TP (g/kg)	TK (m/kg)	NH_4_^+^	NO_3_^−^
S2N0	8.26 ± 0.07 ab	53.26 ± 4.53 a	26.60 ± 2.91 a	188.24 ± 17.84 b	1.73 ± 0.10 a	1.53 ± 0.06 a	0.48 ± 0.01 a	16.06 ± 0.30 a	16.64 ± 1.73 a	3.06 ± 1.14 a
S2N1	8.34 ± 0.04 a	63.24 ± 5.05 a	28.60 ± 4.58 a	243.21 ± 2.92 a	1.78 ± 0.15 a	1.74 ± 0.09 ab	0.50 ± 0.06 a	17.54 ± 6.23 a	17.12 ± 1.23 a	2.72 ± 0.26 a
S2N2	8.25 ± 0.05 ab	77.03 ± 70.06 a	29.55 ± 2.09 a	192.27 ± 25.71 b	1.96 ± 0.11 a	2.01 ± 0.09 ab	0.51 ± 0.01 a	19.90 ± 0.90 a	20.24 ± 3.17 a	2.85 ± 0.89 a
S2N3	8.24 ± 0.06 ab	87.13 ± 7.45 a	30.74 ± 3.51 a	204.75 ± 12.65 b	2.13 ± 0.15 a	1.91 ± 0.07 b	0.52 ± 0.01 a	20.11 ± 0.18 a	20.21 ± 0.87 a	3.24 ± 0.44 a
S2N4	8.19 ± 0.11 b	80.16 ± 7.47 a	29.04 ± 3.91 a	196.50 ± 15.55 b	2.04 ± 0.50 a	2.12 ± 0.08 ab	0.48 ± 0.03 a	19.90 ± 0.54 a	19.95 ± 2.87 a	2.93 ± 0.32a

Note: AN indicates available nitrogen; AP indicates available phosphorus; AK indicates available potassium; SOC indicates soil organic carbon; TN indicates total nitrogen; TP indicates total phosphorus; TK indicates total potassium; NH_4_^+^ indicates ammonium nitrogen; and NO_3_^−^ indicates nitrate nitrogen. Different lowercases indicate statistical significance identified by the Duncan test at *p* < 0.05.

**Table 3 microorganisms-13-00079-t003:** Correlations of soil physico-chemical properties and soil microbial diversity.

	Shannon	Simpson	ACE	Chao1
pH	0.03	−0.25	0.57 *	0.57 *
AN	0.29	0.23	0.23	0.23
AP	−0.05	−0.27	0.20	0.20
AK	−0.20	−0.14	0.47	0.47
SOC	−0.11	0.17	−0.33	−0.33
TN	−0.10	−0.22	−0.03	−0.03
TP	0.17	−0.03	0.67 **	0.67 **
TK	−0.07	0.00	0.01	0.01
NH4	0.21	0.42	−0.04	−0.04
NO3	−0.63 *	−0.48	−0.20	−0.20

* indicates significance at a *p* < 0.05 level, and ** indicates significance at a *p* < 0.01 level.

## Data Availability

Certain proprietary or sensitive data that were generated during the course of this research are not publicly available due to confidentiality agreements or ethical considerations. However, these data are available from the corresponding author upon reasonable request and with permission from the relevant authorities. Interested researchers are invited to contact Sun Yu (email: syfx19801979@163.com) to inquire about accessing these non-public datasets. All data requests will be carefully considered and reviewed to ensure compliance with all applicable laws, ethical guidelines, and data use agreements. This Data Availability Statement serves as a transparent link between the findings presented in this article and the underlying evidence, adhering to Springer Nature’s commitment to research transparency and data accessibility.

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
