# Peer review of "Effects of Different Nitrogen Fertilizer Application Rates on Soil Microbial Structure in Paddy Soil When Combined with Rice Straw Return"

_microorganisms, 2025, doi:10.3390/microorganisms13010079_

Round 1
Reviewer 1 Report
Comments and Suggestions for Authors
Explain concisely why you used a single amount of straw return (7500 kg/hm2); substrate composition? Decomposition rate?
Why did you not include a crop to evaluate its yield with these treatments? It would be very interesting.
Author Response
Dear Assistant Editor Ms. Neeranuch Rukying,
Dear Reviewer 1,
Thank you for your letter and for the comments concerning our manuscript entitled “Effects of Different Nitrogen Fertilizer Application Rates on Soil Microbial Structure in Paddy Soil When Combined with Rice Straw Return (microorganisms-3288980)”. Those comments are all valuable and very helpful for revising and improving our paper. We have studied all provided comments carefully and have made appropriate corrections which we hope meet with approval. The corrections made in the paper and the respective responses to your comments are listed below and shown by revision format in the improved version of the text.
Reviewer1
Comment
Explain concisely why you used a single amount of straw return (7500 kg/hm2); substrate composition? Decomposition rate?
Why did you not include a crop to evaluate its yield with these treatments? It would be very interesting.
Response:
We have added an explanation for this question, please see the M&M and follow:
“Amount of straw returned to the field (see table 1, 7500 kg/hm2) is calculated according to the actual local production needs. The grain-to-grass ratio of the straw returned to the field is an important parameter for determining the amount of rice straw returned to the field. It helps evaluate the relationship between the yield of straw and the grain yield, and then determines the reasonable amount of straw returned to the field (this experiment is calculated based on a grain-to-grass ratio of 1:1).”
Since the article mainly focuses on the soil microbial structure of paddy fields and the direct impact of straw return on soil properties, substrate composition, decomposition rate and yield are not provided.
Reviewer 2 Report
Comments and Suggestions for Authors
This manuscript is within the scope of the journal as adel with microbial comunities in agricultural soil regarding application of different N doses with straw return.
Line 46: please put short notice why this decomposition is skow, possible soil N depression
I hope there are lots of studies with straw usage in production, please find some, at least some meta analyses.
Line 82: more common is to use ha than hm2
Materials: you started with experiment in 2017, but you collected soil samples in 2023, did you have 5-6 years of studies in row or what, thsi is not clear?
What type of soil it was used, please put main physiochemical properties of it.
What type of fertilizers you use, maybe with that answer you can explain some results and improve disscusion
Please put Table 2. in Results section and explain results you got with soil analyses
Explain in captions of Table2 what means abbreviatins as AN, AP…. And what are the units values are expressed
Disscusion organize in the way as you have results, if you put first dana on soil fertility in Results please
I will not put significance values in Conclusion.
Author Response
Dear Assistant Editor Ms. Neeranuch Rukying,
Dear Reviewer 2,
Thank you for your letter and for the comments concerning our manuscript entitled “Effects of Different Nitrogen Fertilizer Application Rates on Soil Microbial Structure in Paddy Soil When Combined with Rice Straw Return (microorganisms-3288980)”. Those comments are all valuable and very helpful for revising and improving our paper. We have studied all provided comments carefully and have made appropriate corrections which we hope meet with approval. The corrections made in the paper and the respective responses to your comments are listed below and shown by revision format in the improved version of the text.
Comment:
This manuscript is within the scope of the journal as adel with microbial comunities in agricultural soil regarding application of different N doses with straw return.
Line 46: please put short notice why this decomposition is skow, possible soil N depression
I hope there are lots of studies with straw usage in production, please find some, at least some meta analyses.
Response:
Dear editor,
Thank you for your considerable comment. We revised our manuscript according to your suggestion. Please see follow:
“When rice straw is introduced into farmland, it undergoes decomposition facili-tated by soil microorganisms, releasing essential nutrients like nitrogen (N), phosphorus (P), and potassium (K) for crop utilization [5,6]. This process helps to decrease the reli-ance on chemical fertilizers, thereby enhancing economic benefits and improving the production-to-input ratio [7,8]. However, upon being submerged in water and decom-posing in the fields, rice straw may release toxins that could potentially harm rice growth. Additionally, crop straws typically exhibit a high carbon-to-nitrogen ratio, ranging from 60:1 to 80:1, which slows down the decomposition of rice residues in the soil [9,10]. Consequently, microorganisms absorb the nitrogen required for crop growth and development, leading to competition that can initially inhibit rice growth during its early stages. Nonetheless, as the rice plants continue to grow and develop, the decom-posed straw releases nutrients that positively impact rice cultivation [11].
Wang et al. [12] conducted an experiment on incorporating rice straw into fields under no-till conditions, revealing that while the number of water-damaged leaves and dry matter accumulation remained unaffected, the practice extended the functional pe-riod of green leaves post-flowering and enhanced yield formation. Furthermore, it led to an increase in the proportion of photosynthetic product accumulation. Similarly, Meng et al. [13] observed that although straw decomposition initially inhibits rice growth and development, it becomes beneficial in later stages. Additionally, the application of or-ganic fertilizer to white soil rice fields significantly boosts the plant's uptake of nitrogen, phosphorus, and potassium. 14. Fentie et al. [14] further found that the incorporation of water hyacinth dry straw notably increased the absorption of these nutrients during the rice's maturity stage. In summary, numerous scholars have extensively investigated the impact of straw incorporation on crop growth and soil fertilization. However, there is a notable gap in systematic research exploring the changes in soil physicochemical properties and microbial activity under the specific conditions of rice straw incorpora-tion in Heilongjiang Province.”
Line 82: more common is to use ha than hm2
Response:
We revised our entire manuscript according to your comment. Please see the revision manuscript.
Materials: you started with experiment in 2017, but you collected soil samples in 2023, did you have 5-6 years of studies in row or what, thsi is not clear?
Response:
Thank you for your considerable comment. We revised our manuscript according to your comment. Please see follow and M&M:
“The research site, situated in a temperate zone characterized by a cold climate, was selected to comprehensively assess the impact of long-term straw incorporation and fer-tilizer application on soil properties and crop growth. To this end, soil samples were diligently collected in 2023, following six years of consistent practice. ”.
What type of soil it was used, please put main physiochemical properties of it.
Response:
Thank you for your considerable comment. We revised our manuscript according to your comment and please see follow and M&M:
“The soil type utilized for this test is black soil. The nutrient content of the 0-20 cm soil layer is detailed as follows: the soil organic matter content is 26.50 g/kg, the total nitrogen content is 2.01 g/kg, the effective nitrogen content stands at 79.56 mg/kg, the effective phosphorus content is 55.84 mg/kg, and the available potassium content is 168.42 mg/kg.”.
What type of fertilizers you use, maybe with that answer you can explain some results and improve disscusion
Response:
We revised it in the M&M. see them in the M&M and follow:
“Urea served as the nitrogen fertilizer in this study, and it was administered in three separate applications: as a basal fertilizer, a tillering fertilizer, and a greening fertilizer. The total amounts of nitrogen fertilizer (in terms of pure nitrogen) applied were as follows: (N4) 173 kg•ha, (N3) 133 kg•ha, (N2) 93 kg•ha, (N1) 53 kg•ha, and (N0) 0 kg•ha. For all treatments, a consistent amount of phosphorus and potassium was applied, with 46 kg•ha of pure phosphorus and 75 kg•ha of pure potassium, respectively. Diammonium phosphate was chosen as the phosphate fertilizer and was applied once as a basal fertilizer. Potassium chloride was used as the potassium fertilizer and was administered in two applications: as a basal fertilizer and as a tillering fertilizer.”
Please put Table 2. in Results section and explain results you got with soil analyses
Response: we revised our manuscript according to your comment. Please see the section of result and follow:
“Under the practice of straw incorporation, varying quantities of chemical fertilizers ex-erted a significant influence solely on soil pH, total nitrogen, total soil phosphorus, and soil available potassium (as indicated in Table 2, with P<0.05). However, no notable effect was observed on other soil nutrient contents (as depicted in Table 2, with P>0.05). Nota-bly, soil total nitrogen exhibited a pattern of initial decrease followed by an increase as the application rate of chemical fertilizers diminished, achieving its peak in the N4 treatment. Specifically, the total nitrogen content in the N4 treatment surpassed that in the N0 treatment by a significant margin of 17.61% (as shown in Table 2, P<0.05). Con-versely, soil total phosphorus demonstrated an initial increase and subsequent decrease with decreasing chemical fertilizer application, peaking in the N2 treatment. Regarding soil available potassium content, the trend was observed to be N3>N1>N0>N2>N4, with the N4 treatment showing a significant difference compared to the other four treatments.”.
Explain in captions of Table2 what means abbreviatins as AN, AP…. And what are the units values are expressed
Response: we revised our manuscript according to your comment. Please see the section of result and follow:
“Note: AN indicated available Nitrogen; AP indicated Available Phosphorus; AK indi-cated Available Potassium; SOC indicated Soil Organic Carbon; TN indicated Total Ni-trogen; TP indicated Total Phosphorus; TK indicated Total Potassium; NH4+ indicated Ammonium Nitrogen; NO3- indicated Nitrate Nitrogen.”.
Disscusion organize in the way as you have results, if you put first dana on soil fertility in Results please
I will not put significance values in Conclusion.
Response: we have revised our conclusion according to your suggestion and please see them.
Reviewer 3 Report
Comments and Suggestions for Authors
Microorganisms
Manuscript Draft
Manuscript Number: 3288980
Title: Effects of Different Nitrogen Fertilizer Application Rates on Soil Microbial Structure in Paddy Soil When Combined with Rice Straw Return
Article Type: Research article
General Comments on MDPI Questions that Reviewers must answer:
- Is the manuscript clear, relevant for the field and presented in a well-structured manner?
The manuscript is written relatively clearly, is well-structured, and is relevant to the field since it analyzes the impact on soil microbial communities in rice cropping ago-ecosystems with different levels of nitrogen fertilizer application keeping rice straw return and phosphorus fertilization constant. Given the potential contribution of this research to the general understanding of the impacts of conventional agricultural fertilization on soil microbial diversity in agricultural systems, this manuscript warrants publication. However, there needs to be TWO substantive corrections made to the manuscript:
1) Please add another sub-section 4.4. added to the Discussion section on how the research results support past research results on changes to alpha-diversity with increasing levels of nitrogen application within agro-ecosystems. You will cite more references in this section to bring the total citations to around 50. For example, please see work of Dr. Tilman at University of Minnesota on this topic and branch out from there. This will provide better context of your research results.
2) Please in the Results section include in table/figure format the results from the diversity indicators that were calculated.
In addition to this one major substantive edit, there are minor edits and clarifications need to be made as well:
3) On L5, please do not use all capital letters for co-authors.
4) In the Abstract, please do NOT use abbreviations.
5) On L33, the first keyword word is capitalized. Also, the keywords need to be in alphabetical order.
6) In Table 1, the header Treat- ment should use a hyphen to split the word to fit the column width. Please use this process throughout the manuscript tables. Also, Table 2 does not have a bold header for the first column.
7) For Table 2 below L121, please do NOT use +/- since you are using abc significant differences. It will make the data more easily understood and more clear.
8) Throughout the manuscript write out as Figure and NOT Fig.
9) For Figures, please stack images on top and make bigger so the writing is readable. Also, increase the font size of the image. They all are too small and it is impossible to read (with the exception of Figure 5 and Figure 6).
10) All headers and sub-headers in the Discussion section need to be 4. Discussion for example.
11) Break up L232-259 into shorter paragraphs.
12) Change to 5. Conclusions on L315.
13) Please add at a couple of sentences at the end of the Conclusions paragraph on how future research can expand upon the current work.
14) On L329-333, use initials for example S.F. and not fully written out names in all caps.
15) Please refer back to the MDPI Microorganisms Word template for authors and make sure all Back Matter sections between the Conclusions and References are added. Some are written as Not applicable for example but they all have to be there (aside from Supplementary Materials since there are none).
16) In the References, the year needs to be in bold without parentheses and moved after the abbreviated journal name making sure to follow the year by a comma. Also, the abbreviated journal name needs to be in italics. Also, author names are not to be written in all caps.
- Are the cited references mostly recent publications (within the last 5 years) and relevant? Does it include an excessive number of self-citations?
About half of the cited references have been published within the last 5 years and appear relevant to the research topic. There are no excessive self-citations.
- Is the manuscript scientifically sound and is the experimental design appropriate to test the hypothesis?
The manuscript is scientifically sound and the experimental analyses are appropriate.
- Are the manuscript’s results reproducible based on the details given in the methods section?
The manuscript’s experimental results are reproducible based on what is described in 2. Materials and Methods.
- Are the figures/tables/images/schemes appropriate? Do they properly show the data? Are they easy to interpret and understand? Is the data interpreted appropriately and consistently throughout the manuscript? Please include details regarding the statistical analysis or data acquired from specific databases.
Please see edits for tables and figures. It is surprising that there is no experiment results of diversity indicators that were calculated presented in table(s) nor figure(s).
- Are the conclusions consistent with the evidence and arguments presented?
The Conclusions are consistent with the evidence and arguments presented. Please add a couple of sentences on how future research can improve upon the current work at the end of the Conclusions section.
- Please evaluate the data availability statements to ensure it is adequate.
Please add the Data Availability Statement. The Ethics Statement is OK.
Author Response
Dear Assistant Editor Ms. Neeranuch Rukying,
Dear Reviewer 3,
Thank you for your letter and for the comments concerning our manuscript entitled “Effects of Different Nitrogen Fertilizer Application Rates on Soil Microbial Structure in Paddy Soil When Combined with Rice Straw Return (microorganisms-3288980)”. Those comments are all valuable and very helpful for revising and improving our paper. We have studied all provided comments carefully and have made appropriate corrections which we hope meet with approval. The corrections made in the paper and the respective responses to your comments are listed below and shown by revision format in the improved version of the text.
Reviewer3
Comments:
Microorganisms
Manuscript Draft
Manuscript Number: 3288980
Title: Effects of Different Nitrogen Fertilizer Application Rates on Soil Microbial Structure in Paddy Soil When Combined with Rice Straw Return
Article Type: Research article
General Comments on MDPI Questions that Reviewers must answer:
- Is the manuscript clear, relevant for the field and presented in a well-structured manner?
The manuscript is written relatively clearly, is well-structured, and is relevant to the field since it analyzes the impact on soil microbial communities in rice cropping ago-ecosystems with different levels of nitrogen fertilizer application keeping rice straw return and phosphorus fertilization constant. Given the potential contribution of this research to the general understanding of the impacts of conventional agricultural fertilization on soil microbial diversity in agricultural systems, this manuscript warrants publication. However, there needs to be TWO substantive corrections made to the manuscript:
1) Please add another sub-section 4.4. added to the Discussion section on how the research results support past research results on changes to alpha-diversity with increasing levels of nitrogen application within agro-ecosystems. You will cite more references in this section to bring the total citations to around 50. For example, please see work of Dr. Tilman at University of Minnesota on this topic and branch out from there. This will provide better context of your research results.
Response:
Thank you for your suggestion and we revised our manuscript according to your suggestion. Please see the section of discussion.
2) Please in the Results section include in table/figure format the results from the diversity indicators that were calculated.
Response:
We revised our results section carefully according to your suggestion. Please see the result section.
In addition to this one major substantive edit, there are minor edits and clarifications need to be made as well:
3) On L5, please do not use all capital letters for co-authors.
Response: we revised this error. Thank you.
4) In the Abstract, please do NOT use abbreviations.
Response: we revised this error. Thank you.
5) On L33, the first keyword word is capitalized. Also, the keywords need to be in alphabetical order.
Response:
We revised our manuscript according to your suggestion. Please see the keywords.
6) In Table 1, the header Treat- ment should use a hyphen to split the word to fit the column width. Please use this process throughout the manuscript tables. Also, Table 2 does not have a bold header for the first column.
Response:
We revised our entire manuscript according to your comments. Please see our revision manuscript.
7) For Table 2 below L121, please do NOT use +/- since you are using abc significant differences. It will make the data more easily understood and more clear.
Response:
We revised our entire manuscript according to your comments. Please see our revision manuscript.
8) Throughout the manuscript write out as Figure and NOT Fig.
Response:
We revised our entire manuscript according to your comments. Please see our revision manuscript.
9) For Figures, please stack images on top and make bigger so the writing is readable. Also, increase the font size of the image. They all are too small and it is impossible to read (with the exception of Figure 5 and Figure 6).
Response:
We adjust our entire figures according to your comments. Please see our revision manuscript.
10) All headers and sub-headers in the Discussion section need to be 4. Discussion for example.
Response:
We revised our entire Discussion of manuscript according to your comments. Please see our revision manuscript.
11) Break up L232-259 into shorter paragraphs.
Response:
We revised this paragraphs according to your comments. Please see our revision manuscript.
12) Change to 5. Conclusions on L315.
Response:
We revised this error.
13) Please add at a couple of sentences at the end of the Conclusions paragraph on how future research can expand upon the current work.
Response:
We revised our conclusion according to your comment. Please see the conclusion and follow:
“This practice has a favorable effect on preserving the health and stability of the soil.”.
14) On L329-333, use initials for example S.F. and not fully written out names in all caps.
Response: we revised these errors according to your comment.
15) Please refer back to the MDPI Microorganisms Word template for authors and make sure all Back Matter sections between the Conclusions and References are added. Some are written as Not applicable for example but they all have to be there (aside from Supplementary Materials since there are none).
Response:
Thank you for your suggestion. We revised our manuscript according to your suggestion.
16) In the References, the year needs to be in bold without parentheses and moved after the abbreviated journal name making sure to follow the year by a comma. Also, the abbreviated journal name needs to be in italics. Also, author names are not to be written in all caps.
Response:
Thank you for your suggestion. We revised our manuscript according to your suggestion.
- Are the cited references mostly recent publications (within the last 5 years) and relevant? Does it include an excessive number of self-citations?
About half of the cited references have been published within the last 5 years and appear relevant to the research topic. There are no excessive self-citations.
- Is the manuscript scientifically sound and is the experimental design appropriate to test the hypothesis?
The manuscript is scientifically sound and the experimental analyses are appropriate.
- Are the manuscript’s results reproducible based on the details given in the methods section?
The manuscript’s experimental results are reproducible based on what is described in 2. Materials and Methods.
- Are the figures/tables/images/schemes appropriate? Do they properly show the data? Are they easy to interpret and understand? Is the data interpreted appropriately and consistently throughout the manuscript? Please include details regarding the statistical analysis or data acquired from specific databases.
Please see edits for tables and figures. It is surprising that there is no experiment results of diversity indicators that were calculated presented in table(s) nor figure(s).
Response:
We revised our the result according to your suggestion. Please see section of result.
- Are the conclusions consistent with the evidence and arguments presented?
The Conclusions are consistent with the evidence and arguments presented. Please add a couple of sentences on how future research can improve upon the current work at the end of the Conclusions section.
Response:
We revised our conclusion according to your comment. Please see the conclusion and follow:
“This practice has a favorable effect on preserving the health and stability of the soil.”.
- Please evaluate the data availability statements to ensure it is adequate.
Please add the Data Availability Statement. The Ethics Statement is OK.
Response: we add the Data Availability Statement in the revision manuscript.
Reviewer 4 Report
Comments and Suggestions for Authors
The manuscript "Effects of different nitrogen fertilizer application rates on soil microbial structure in paddy soil when combined with rice straw return" present an important research for the field of soil microbiome. The analysis of fertilizer-straw-microbiome interaction is necessary to increase the knowledge on these processes and interactions, and to provide a solid base for new microbial applications.
The Abstract is clear and informative, presenting the main findings of this study.
The Introduction offers the background information and sustain the necessity of this study.
The aim and the objectives are clear presented.
Materials and Methods are explicit and provide all the necessary information for the replication of the experiment and the understanding of the results obtained.
Results section - a suggestion for this section - expand the results explanation and interpretation. The figures are clear and informative, but they should be completely explored. This will sustain better the Discussion section.
Discussion - All the three sub-sections of Discussion expand the interpretation of the observed trends and sustain the observations from this research. The observations are linked to international literature in the field.
Conclusion section - this section present in a condensed form the results obtained.
Overall, the manuscript is interesting and brings new information on soil microbiome.
Author Response
Dear Assistant Editor Ms. Neeranuch Rukying,
Dear Reviewer 4,
Thank you for your letter and for the comments concerning our manuscript entitled “Effects of Different Nitrogen Fertilizer Application Rates on Soil Microbial Structure in Paddy Soil When Combined with Rice Straw Return (microorganisms-3288980)”. Those comments are all valuable and very helpful for revising and improving our paper. We have studied all provided comments carefully and have made appropriate corrections which we hope meet with approval. The corrections made in the paper and the respective responses to your comments are listed below and shown by revision format in the improved version of the text.
Reviewer 4,
The manuscript "Effects of different nitrogen fertilizer application rates on soil microbial structure in paddy soil when combined with rice straw return" present an important research for the field of soil microbiome. The analysis of fertilizer-straw-microbiome interaction is necessary to increase the knowledge on these processes and interactions, and to provide a solid base for new microbial applications.
The Abstract is clear and informative, presenting the main findings of this study.
The Introduction offers the background information and sustain the necessity of this study.
The aim and the objectives are clear presented.
Materials and Methods are explicit and provide all the necessary information for the replication of the experiment and the understanding of the results obtained.
Results section - a suggestion for this section - expand the results explanation and interpretation. The figures are clear and informative, but they should be completely explored. This will sustain better the Discussion section.
Response:
We revised our manuscript according to you and other three reviewers comment. Please see the revision manuscript.
Discussion - All the three sub-sections of Discussion expand the interpretation of the observed trends and sustain the observations from this research. The observations are linked to international literature in the field.
Response:
We revised our manuscript according to you and other three reviewers comment. Please see the revision manuscript.
Conclusion section - this section present in a condensed form the results obtained.
Overall, the manuscript is interesting and brings new information on soil microbiome.